# Fatty Acid Synthase as Interacting Anticancer Target of the Terpenoid Myrianthic Acid Disclosed by MS-Based Proteomics Approaches

**DOI:** 10.3390/ijms25115918

**Published:** 2024-05-29

**Authors:** Alessandra Capuano, Gilda D’Urso, Erica Gazzillo, Gianluigi Lauro, Maria Giovanna Chini, Maria Valeria D’Auria, Maria Grazia Ferraro, Federica Iazzetti, Carlo Irace, Giuseppe Bifulco, Agostino Casapullo

**Affiliations:** 1Department of Pharmacy, University of Salerno, 84084 Fisciano, Italy; acapuano@unisa.it (A.C.); egazzillo@unisa.it (E.G.); glauro@unisa.it (G.L.); bifulco@unisa.it (G.B.); casapullo@unisa.it (A.C.); 2PhD Program in Drug Discovery and Development, Department of Pharmacy, University of Salerno, 84084 Fisciano, Italy; 3Department of Biosciences and Territory, University of Molise, C.da Fonte Lappone, 86090 Pesche, Italy; 4Department of Pharmacy, University of Naples “Federico II”, Via Domenico Montesano 49, 80131 Naples, Italy; mariavaleria.dauria@unina.it; 5Department of Molecular Medicine and Medical Biotechnology, University of Naples Federico II, 80131 Naples, Italy; mariagrazia.ferraro@unina.it; 6Biochem Lab, Department of Pharmacy, School of Medicine and Surgery, University of Naples “Federico II”, Via Domenico Montesano 49, 80131 Naples, Italy; federica.iazzetti@unina.it (F.I.); carlo.irace@unina.it (C.I.)

**Keywords:** drug discovery, functional proteomics, fatty acid synthase, t-LiP, DARTS, molecular docking, preclinical investigations

## Abstract

This research focuses on the target deconvolution of the natural compound myrianthic acid, a triterpenoid characterized by an ursane skeleton isolated from the roots of *Myrianthus arboreus* and from *Oenothera maritima* Nutt. (Onagraceae), using MS-based chemical proteomic techniques. Application of drug affinity responsive target stability (DARTS) and targeted-limited proteolysis coupled to mass spectrometry (t-LiP-MS) led to the identification of the enzyme fatty acid synthase (FAS) as an interesting macromolecular counterpart of myrianthic acid. This result, confirmed by comparison with the natural ursolic acid, was thoroughly investigated and validated in silico by molecular docking, which gave a precise picture of the interactions in the MA/FAS complex. Moreover, biological assays showcased the inhibitory activity of myrianthic acid against the FAS enzyme, most likely related to its antiproliferative activity towards tumor cells. Given the significance of FAS in specific pathologies, especially cancer, the myrianthic acid structural moieties could serve as a promising reference point to start the potential development of innovative approaches in therapy.

## 1. Introduction

Target deconvolution implies the identification of the biological targets of a small molecule within the cell’s content [1]. This is essential for understanding the biological and pharmacological properties of small molecules, like drugs and the knowledge of disease mechanisms, assessing the potential polypharmacological or toxic effects, and providing crucial information for a rational drug design approach [2,3].

Target deconvolution strategies applied to natural products can reveal the mechanisms underlying their biological effects and uncover new chemical scaffolds able to specifically interact with proteins playing significant roles in pathological states. This exploration helps to discover novel structures with potential applications in the drug development process. 

There are several examples of libraries [4] containing numerous natural products (NPs) belonging to different classes, including alkaloids, polyketides, terpenoids, and flavonoids. Among these, terpenoids constitute the largest class of natural products, contributing to approximately 60% of the overall diversity in natural product compounds [5]. Therefore, meaningful research efforts have been directed at exploring and understanding the scaffolds associated with terpenoid natural products. In particular, triterpenic acids featuring lupane, oleanane, and ursane structural cores, such as betulinic, oleanolic, and ursolic acids, respectively, demonstrate noteworthy anticancer, anti-inflammatory, and hepatoprotective activities. Recent studies have shown that triterpenoids hold promise as agents for treating and inhibiting breast cancer through various molecular mechanisms, including their ability to inhibit angiogenesis and differentiation, regulate apoptosis, inhibit DNA polymerase, alter signal transduction, and hamper metastasis [6,7].

Myrianthic acid (MA) (Figure 1) is a triterpenoid characterized by an ursane skeleton isolated from the roots of *Myrianthus arboreus* [8] and from *Oenothera maritima* Nutt. (Onagraceae) [9]. Interestingly, triterpenoids with the ursane skeleton have been extensively reported for their diverse biological activities, particularly in the areas of antidiabetic [10] and anticancer research [11].

MA demonstrated interesting biological properties, such as antioxidant activity and moderate inhibitory effects on thrombin [9], suggesting it is a candidate for deeper investigations on the specific macromolecular targets involved in such pathways. 

Considering the promising biological MA profile, this project sought to comprehensively explore and identify its profile of interaction with specific targets coming from a human cell proteome. This analysis was accomplished employing a proteomic strategy that combined two mass spectrometry-based limited proteolysis techniques: drug affinity responsive target stability (DARTS) and targeted-limited proteolysis coupled to mass spectrometry (t-LiP-MS). These two approaches exploit the stabilization of a protein structure following its binding with a ligand. Indeed, after this event, the protein structure becomes more compact and, therefore, more resistant to proteolysis, using nonspecific proteases employed under controlled limited conditions. Thanks to this reduced sensitivity to the enzyme action, it is possible to identify the target proteins of a small molecule and identify the specific peptide regions involved in the binding with the small molecule. Specifically, the DARTS technique was employed to identify the potential MA protein targets, whereas t-LiP gave information on the interaction inside the protein-ligand complex. DARTS is a label-free proteomic technique employed in drug discovery to identify potential protein targets for small molecules (SM) [12,13]. In a typical DARTS experiment, a complex biological sample (e.g., cell lysate) containing the target protein is firstly incubated with the SM and treated with a broad specific protease, e.g., subtilisin, under controlled conditions. Proteins are separated in SDS-PAGE electrophoresis, and after gel staining, the single bands are excised from the gel, and proteins are submitted to tryptic digestion for the next bottom-up mass spectrometric analysis. Thanks to the appropriate software, it is possible to compare acquired spectra with those in database libraries. This approach enables the identification of various proteins and allows for a relative quantification of each. By comparing the protease digestion patterns in the presence and absence of the small molecule, it is possible to pinpoint proteins likely interacting with the compound [14,15]. In the present study, DARTS experiments identified some putative protein targets, among which fatty acid synthase (FAS) was selected as the most interesting due to its crucial role in various pathological processes, such as cancer. 

This finding was confirmed by comparison with the natural triterpene analog ursolic acid, which was found to share the same macromolecular target with MA. While in the case of ursolic acid, the previous investigations were more focused on its inhibition profile from a biological point of view, our analyses featured a more analytical and chemical footprint towards the study of the interactions occurring in the MA/FAS complex. 

FAS is a large enzymatic system involved in fatty acid synthesis, comprising seven domains. Although FAS is ubiquitously expressed in all tissues, de novo fatty acid synthesis typically occurs at low levels, with dietary sources meeting the usual demand [16]. However, in various pathological conditions such as diabetes, obesity, and cancer, FAS has been found to be overexpressed. Indeed, overexpression of FAS in specific tumors (e.g., breast, prostate and ovarian tumors) has been correlated with aggressive cancer phenotype and poor prognosis [17]. Fatty acids (FAs) play a pivotal role in lipid metabolism, serving as essential components in the construction of biological membranes and as precursors for secondary messengers. In contrast to normal cells, which primarily rely on external FA uptake, cancer cells adopt a dual strategy, obtaining FAs both from the microenvironment and through FAS-driven de novo synthesis. Current anti-cancer therapy focuses on the pharmacological inhibition of FAS, with specific strategies able to selectively act on diverse malignant cells, whether in vitro and in vivo, while sparing normal cells from the inhibitory effects. Given the importance of this enzyme, our following studies focused on a deep analysis of the FAS-MA complex. t-LiP experiments enabled the identification of the protein regions involved in the binding with MA, and these findings were further confirmed through molecular docking studies. The following biological assays evaluated the MA cytotoxic activity against selected preclinical human tumor models overexpressing FAS (i.e., the MCF-7 estrogen-responsive breast adenocarcinoma and the MDA-MB-231 triple-negative breast adenocarcinoma) and confirmed the MA inhibitory action on the FAS enzymatic activity. This study highlights the promising role that MA can play in drug discovery by revealing a new mechanism of action that can be leveraged in the search for novel therapeutic compounds.

## 2. Results

### 2.1. DARTS for Myrianthic Acid Target(s) Identification

To investigate the interaction profile of MA, DARTS experiments were conducted using a lysate of HeLa tumor cells. The protein extract was then aliquoted, and various amounts of MA dissolved in DMSO (1 μM, 10 μM, 100 μM) were added to the different aliquots, keeping one aliquot as a reference, adding to this sample only the vehicle.

A different amount of subtilisin was added to each aliquot, performing a limited proteolysis step, followed by SDS-PAGE electrophoresis. After Coomassie staining, the gel was inspected to highlight protein bands whose intensity increased proportionally to MA concentrations. Two bands meeting these criteria were identified, corresponding to molecular weights close to 235 and 90 kDa, respectively (Appendix A).

Thus, these bands were carefully excised from the gel and submitted to an in situ tryptic digestion [18] to give peptide mixtures for the following nano-ESI-LC-MS/MS analysis, performed on a nano-flow UPLC system interfaced with the nano-ESI source of a Themo Q-Exactive mass spectrometer. 

The obtained data underwent initial processing through the Xcalibur software 2.2 version to produce peak lists containing extensive MS information about the peptide mixtures. Subsequently, these lists were examined against the SwissProt Human database using the Proteome Discoverer (PD) bioinformatics server. This system matches experimental data with those reported in SwissProt database, facilitating protein identifications. Additionally, PD allowed for relative quantification, comparing the protein abundances identified in the MA-treated samples with those in the reference sample. Proteins exhibiting higher abundance in the MA-treated samples were considered potential targets.

As illustrated in Table 1, FAS exhibited a good concentration-dependent increase in the abundance ratios and was identified as the most promising target, due to the good reproducibility of the results across three replicates (Table 1 and Appendix A).

To further validate the mass-spectrometric DARTS data, the same DARTS samples underwent Western blotting analysis utilizing anti-FAS (Figure 2). GAPDH (Glyceraldehyde 3-Phosphate Dehydrogenase) served as a loading normalizer, since it is a well-known protein stable against protease digestion, for the subsequent densitometric analysis performed with ImageJ software 1.52a.

### 2.2. Analysis of MA/FAS Interaction Features through t-LiP

After identifying FAS as the most promising cellular partner for MA, we proceeded to characterize the interaction features of their complexes through t-LiP experiments.

t-LiP is a valid technique for studying the ligand–protein complex [19,20]. This method once again leverages the increased protein compactness resulting from ligand interaction, which reduces the accessibility of a proteolytic enzyme to the protein structure. This analysis is focused on the study of the tryptic peptides of the target protein. Since trypsin is a specific enzyme, it is easy to predict the tryptic peptides that can be generated during digestion. These peptides are selectively examined using the multiple reaction monitoring mass spectrometry (MRM-MS) mode based on their *m*/*z* and fragmentation pattern [21,22]. Therefore, this technique allows the recognition of a specific peptide within a complex protein mixture, such as a tryptic-digested lysate. Nevertheless, when trypsin digestion follows a limited proteolysis procedure, some of the tryptic peptides may no longer be detectable as they have previously undergone cleavage by the nonspecific protease. However, if the ligand has protected against limited proteolysis, the nonspecific cleavage sites will not be recognized, and the tryptic peptides will be detectable once again. So, the analysis involves performing a relative quantification between the tryptic peptides of a sample treated with a ligand and a control sample. Peptides exhibiting an increase in abundance are taken as protected by the compound and, therefore, are considered involved in the binding with it.

Before proceeding with the experiment of the FAS-MA complex, computational analysis was required to create the MRM methods for mass spectrometric analysis containing transitions of tryptic peptides (“transitions”: peptide ion and fragment ion pair). This research was conducted using PeptideAtlas Human build database to collect information about the FAS tryptic peptides. In a second moment, for each tryptic peptide, the *m*/*z* values of the most probable fragments were gathered from the Complete Human SRMAtlas build. 

Based on this analysis, a preliminary method containing 95 FAS peptides and their three best fragments, chosen according to their reported ranking score, was built. This method was tested on a HeLa tryptic digested lysate, and just one transition for each peptide was selected, leading to a final MRM method containing 92 transitions (Appendix A). 

For the t-LiP-MRM experiment, HeLa cell lysate was prepared in non-denaturing conditions and incubated with 100 μM MA or DMSO for one hour at room temperature. Samples were then subjected to limited proteolysis with subtilisin under native conditions and then shifted to denaturing conditions for extensive in-solution tryptic digestion, producing peptide mixtures suitable for LC-MRM-MS analysis using the previously optimized MRM method.

The transitions were then examined, and their intensities were compared across the MA-treated sample and control sample. Ten protected FAS peptides were identified, showing a fold change greater than 1.5 (i.e., the ratio between the peak area in the treated sample and that in the control sample) (Table 2).

### 2.3. MA/FAS Complex: Molecular Docking

Computational studies were conducted to elucidate the mechanism of action of MA toward FAS. In more detail, ursolic acid, featuring high similarities with MA, was reported in the literature as an inhibitor of the FAS-MAT domain, one of the different domains of this enzyme [23,24]. Indeed, the overall FAS structure can be divided into two main parts: the lower condensing part, which contains the condensing domains ketoacyl synthase (KS) and malonyl-acetyl transferase (MAT), responsible for the early stages of fatty acid synthesis. The upper modification part includes the dehydratase (DH), the enoyl reductase (ER), and the ketoreductase (KR) domains involved in modifying the fatty acid chain. There are also two additional non-enzymatic domains known as “pseudo-methyltransferase” (ΨME) and “pseudo-ketoreductase” (ΨKR), positioned peripherally [25]. Additionally, the acyl carrier protein (ACP) domain is involved in transporting acyl intermediates during the fatty acid synthesis reaction, while the hydrolysis occurs via the thioesterase (TE) domain that releases palmitic acid [26].

With reference to literature findings, molecular docking experiments were performed to reproduce the possible interactions of both MA and Ursolic acid with FAS. In more detail, only three human crystal structures of the MAT domain were available, namely 2JFD [27], 2JFK [28], and 3HHD [29], in which no co-crystallized substrate and/or inhibitor were reported. Furthermore, the crystal structure of the MAT domain of FAS from mice in complex with its physiological substrate Malonyl Coa (PDB code: 5MY0 [30]) was reported, featuring a percentage of identity = 87.07% in 852 residues (Appendix A). In addition, in this crystal structure, due to the presence of the endogenous substrate, the Met499, which has a role concerning either blocking or enabling the access of substrate to the region of the protein involved in the catalytic activity, is in a conformation that, differently from the human structures, allows the accommodation of putative inhibitors towards the catalytic triad, i.e., His683, Ser581, and Arg606, thus highlighting the protein conformation in 5MY0 as the active one [25]. In fact, the acid group of the Malonyl CoA reported in the 5MY0 is able to establish interaction with the amino acids of the catalytic triad. For these reasons, the protein structure deposited on the PDB with the code 5MY0 was considered for computational investigations. After performing molecular docking experiments with the case-study compounds, the results highlighted that both MA and ursolic acid were able to interact with the fundamental amino acids of MAT domain binding site (Met499, Ser581), giving the chance to outline two possible binding hypotheses. In the first one, the hydroxyl group of ring A interacts with the catalytic triad (Figure 3). In the second hypothesis, the carboxylic moiety interacts with the three fundamental amino acids (Figure 4). These computational outcomes corroborate the possible interaction of MA with the MAT domain of FAS.

### 2.4. In Vitro Bioscreens and IC_50_ Evaluation

The biological activity of MA and its possible antiproliferative action were investigated in specific tumor phenotypes of breast cancer (BC) overexpressing FAS. In particular, we performed preclinical experiments in well-established human BC cell models represented by the estrogen-responsive (ER) adenocarcinoma cells (MCF-7 line) and the triple-negative adenocarcinoma cells MDA-MB-231. The parallel use of human healthy cells—i.e., primary adult dermal fibroblasts (HDFa) and normal breast epithelial cells (MCF-10A)—allowed a preliminary evaluation of the selectivity of action, as well as the exploration of possible cytotoxic effects of MA in non-tumor cells. Under the same experimental conditions, the in vitro models selected for this study were also treated with ursolic acid (UA) and orlistat as reference drugs. Indeed, current research suggested the natural triterpene analogue UA to be endowed with anticancer activity in several human cancer cells, including BC, with an important FAS inhibitory activity [23,31]. Similarly, Orlistat, a semisynthetic derivative of a naturally occurring lipase inhibitor produced by *Streptomyces toxytricini*, has been found to inhibit FAS, thereby displaying anti-tumor properties [32,33]. Overall, the evaluation of cellular responses following in vitro treatments has yielded intriguing results. The cell survival index arising from the concentration–effect curves demonstrates a significant antiproliferative activity of MA in tumor cells, both concentration and time-dependent (see Figure 5). Its biological activity is comparable although less evident than that of UA and Orlistat. However, as shown in Table 3 and Table 4, the IC_50_ values calculated for MA after 48 and 72 h of in vitro treatment are in the low micromolar range for both the MCF-7 cells and the triple-negative MDA-MB-231 phenotype, and therefore are indicative of moderate antiproliferative activity. Conversely, no significant biological effects were observed at the 48 h time point on healthy cultures treated with MA (Figure 6), suggestive of a selectivity of action on cancer cells and a likely good safety profile on normal cells. In fact, even at the highest concentrations (50 µM) and after long exposure times (72 h), only weak interferences with cell growth and proliferation were detected in both MCF-10A and HDFa. MCF-10A cells represent the healthy counterpart of the tumor models used herein and their biological responses to treatments are for this reason of interest; HDFa derives from a primary culture behaving as an ideal model to study cellular responses to in vitro treatments as well as toxicological responses.

Overall, the outcome from in vitro bioscreens in healthy and tumor cells prompts intriguing inquiries regarding the selectivity and targeted properties of MA. These findings suggest its potential as a therapeutic agent, displaying a preference for inducing cytotoxic effects on cancer cells while preserving the viability of normal cells.

### 2.5. Regulation of FAS Activity by Myrianthic Acid

To correlate proteomic-based target identification and in silico results with the antitumor activity highlighted by in vitro experiments, we analyzed the regulation of FAS enzymatic activity by MA in protein extracts obtained from MCF-7 and MDA.MB-231 breast cancer cells. FAS activity was quantified in nanomoles of NADPH oxidized per minute per milligram of protein, and the obtained results were expressed as a percentage (%) of the total FAS activity in the presence of 10 µM of MA, UA, and Orlistat, respectively, as fully reported in the experimental section. A reference point of 100% was assigned to FAS activity in the absence of the considered molecules (Figure 7).

Experiments performed on protein extracts demonstrate that MA is able to significantly inhibit FAS activity. Though the inhibitory trend is very similar, the FAS inhibition is more evident in MCF-7 cell extracts with respect to the triple-negative phenotype. As expected, under the same experimental conditions, the inhibitory effect of UA and Orlistat on FAS activity is even more marked. The experiment was also conducted by directly treating the cancer cell lines with MA, UA, and Orlistat at the 10 µM final concentration for 48 h of incubation. Following in vitro exposure, cells were harvested and then lysed to obtain cell extracts. By setting up the same enzymatic assay, we finally evaluated FAS activity. As shown in Figure 8, the results describing the modulation of the enzymatic activity are very similar to the previous ones and demonstrate that MA is able to significantly inhibit the enzymatic activity of FAS within the cells. The MCF-7 model remains the most sensitive to the biological activity of MA on FAS activity, and these data correlate well with the MA IC_50_ values formerly indicated.

### 2.6. Regulation of FAS Expression by Myrianthic Acid

We finally analyzed the possible regulation of cellular expression of FAS in triple-negative BC cells by Western blot and immunodetection. MDA-MB-231 cells were exposed for 48 h to MA, UA, and Orlistat at a concentration of 10 µM. At the end of the experiment, cells were appropriately collected and lysed to obtain cell extracts, which were subjected to Western blot analysis as described in the experimental section. Immunodetection, depicted in Figure 9, shows a noteworthy modulation of FAS protein cellular levels. In fact, a significant down-regulation of FAS levels was detectable following in vitro treatments of cells with MA and UA. Conversely, incubation with Orlistat did not cause significant changes in FAS expression.

## 3. Discussion

Natural products play a pivotal role in the discovery and development of novel pharmaceuticals. These compounds often possess unique chemical structures and pharmacological properties that make them well-suited for targeting specific biological pathways involved in diseases. Of particular interest are scaffolds associated with triterpenes featuring the ursane structural core, such as ursolic acid, which is known to possess anticancer activity through various molecular mechanisms. In fact, it has been reported to modulate cellular transcription factors, growth factor receptors, inflammatory cytokines, and many other molecular targets that regulate cell proliferation [34].

The present study employed both untargeted and targeted proteomics approaches to explore the interaction profile of myrianthic acid, a natural triterpenoid with notable biological properties.

DARTS experiments were applied to identify the target(s) and revealed fatty acid synthase (FAS) as the most interesting MA protein target. Its protection against limited proteolysis was found to be concentration-dependent with respect to the three MA concentrations used in the experiment, as shown in Table 1. These data were in perfect agreement with those obtained from the Western blot used for further DARTS validation (Figure 2). FAS is a multi-enzyme crucial in various pathological processes, such as different cancer progressions [17,35,36]. It has been known for some decades to be overexpressed in human cancers compared to normal tissues. Indeed, enhanced de novo lipogenesis is believed as a distinctive feature of many tumor phenotypes to support uncontrolled growth and proliferation, where FAS biological activity becomes crucial for the production of metabolic energy as well as for the synthesis of new plasma membranes throughout cell division [37]. Growing evidence now shows FAS as a central player in metabolic rewiring (including carbohydrate and protein metabolism other than the lipid one) in cancer cells with lipogenic phenotypes [38]. In this context, FAS upregulation has also been correlated to the identification of aggressive and metastatic tumor phenotypes [17]. All this makes FAS a pro-oncogenic enzyme and a potential molecular target for cancer therapy, thereby providing an important boost to explore further the biological effects of natural inhibitory molecules such as MA [37].

Confirmation of FAS as a possible cellular target of MA came from a comparison with the triterpene analogue ursolic acid (UA), which was described as FAS inhibitor by Zhang et al., who also reported the inhibition kinetics of UA on FAS [23,31]. The results indicated that UA competitively inhibited FAS against acetyl-CoA and malonyl-CoA but exhibited uncompetitive inhibition against NADPH [24]. Thus, FAS inhibition is already known in therapeutic contexts, particularly in oncology, since cancer cells often exhibit FAS over-expression. The selective inhibition of FAS can interfere with tumor cells’ ability to synthesize the necessary fatty acids for proliferation and survival, and its inhibitors are actively investigated as anticancer agents [35].

On this basis, we started a deep analysis of the interactions between MA and FAS to provide more useful information for the following studies. t-Lip-MS allowed the identification of several protein regions undergoing conformational changes upon MA binding. These modifications made these sites less accessible to the enzyme and, therefore, less hydrolysable. The identified peptide regions (Table 2) were distributed across different domains of the enzyme. The protein regions located in the FAS domains were further confirmed by molecular docking studies using UA as a comparative sample. In fact, in silico studies highlighted two possible binding modes of MA with the MAT domain of the FAS (Figure 3 and Figure 4). Moreover, we supposed that the second hypothesis (Figure 4) was the most robust one due to the precise accommodation of the carboxylic group towards the MAT binding site, which was similar to the carboxylic group of the natural substrate malonyl CoA reported in the respective crystal structure (PDB code: 5MY0) [27,28,29]. In fact, the analysis of the output docking poses disclosed the ability of the carboxylic group of MA to interact with the catalytic triad as well as the malonyl CoA. In addition, a similar behaviour was also detected with the already reported FAS inhibitor, namely the UA, thus further corroborating the inhibitory activity and the ability of myrianthic acid to interact with the MAT domain of FAS [24].

Prompted by proteomic and in silico evidence, biological evaluations were performed to investigate MA cytotoxic activity against tumor cell lines, as well as its inhibitory action on FAS enzymatic activity. Preclinical studies in cellular models of breast cancer highlighted a significant antiproliferative activity of myrianthic acid. This biological activity is comparable to that of UA, which has already been proven to be a promising phytochemical endowed with strong anticancer effects against BC cells both in vitro and in vivo. In particular, ursolic acid and its derivatives appear to inhibit breast cancer proliferation via a variety of molecular mechanisms including cell cycle arrest, regulation of key proteins involved in signal transduction pathways, and intrinsic and extrinsic apoptosis induction by regulation of several anti-apoptotic and pro-apoptotic proteins [31,36]. Given the molecular similarities between UA and MA, it is possible that these molecules can share biomolecular targets and trigger the same programmed cell death pathways in tumor cells. Of note were cytotoxic outcomes obtained in the triple-negative phenotype MDA-MB-231, which is usually refractory to many types of anti-tumor treatments [39]. For this type of breast cancer marked by dismal prognosis, chemotherapy currently remains the only effective option, which is the reason why the search for new and safe chemotherapeutic agents is a foremost objective for researchers [40]. In line with literature data concerning the UA analogue, MA has also shown to be safe and biocompatible in preclinical tests, where it does not exhibit cytotoxic effects on normal cells [36].

As far as enzymatic activity is concerned, the results obtained from this study reveal a noteworthy direct inhibitory effect of myrianthic acid on the FAS enzyme. The observed inhibitory activity is comparable to that exhibited by both the analog ursolic acid and known FAS inhibitor orlistat [23,33]. Our preclinical experiments also demonstrate that MA is able to act as a potential anticancer drug against human BC lines, significantly inhibiting the enzymatic activity of FAS in cells and thus controlling tumor growth in vitro. Interestingly, FAS’s implications in tumorigenesis were first discovered in human breast carcinoma cells, although it was later found to be overexpressed in many tumor phenotypes of different histological origins [41]. Thus, FAS selective regulation by natural compounds with effects on tumor cell proliferation can open new possibilities in the therapeutic field [37]. Moving in this direction, in-depth understanding of enzyme–inhibitor interaction mechanisms could provide new knowledge for the optimization of natural molecular platforms in order to improve their biological activity and establish novel therapeutic protocols.

Interestingly, protein expression studies by immunodetection reveal that MA was also able to decrease the cellular amount of FAS, and this effect was similar to that of UA. Conversely, Orlistat did not affect the expression profile of FAS. It has been reported that natural plant triterpenoids, including UA, can interfere with the control of gene expression of many cellular proteins. According to former data, these kinds of effects have been described both in vitro and in vivo and correlated to lipogenesis and fatty acid oxidation. In particular, among various genes related to the control of lipid metabolism, UA was shown to decrease FAS mRNA expression in mice, as well as to reduce lipid accumulation by suppressing FAS expression in liver cells [42,43]. As well as the inhibition of its enzymatic activity, the decreased cellular levels of FAS could also be concerned with the deregulation of cellular energetics in tumors. Thus, these xenobiotic-induced regulations could have important impacts on the viability of lipogenic tumor phenotypes. However, further investigations are required to outline the biological effect of MA on FAS expression and activity, and to fully explore its possible use as a prospective therapeutic agent targeting FAS protein. Based on experimental evidence, it has also been suggested that UA-induced FAS down-regulation might be caused by activation of the AMPK signalling pathway, which in turn reduces lipogenesis and increases lipolysis [42]. In addition, UA has been found to regulate the expression of lipid metabolism genes via the activation of PPAR-α in vitro [44]. Considering the molecular similarities between these triterpene derivatives, it is reasonable to predict the involvement of the same pathways to explain MA-dependent FAS regulation.

Overall, these findings suggest that MA may serve as a promising hit compound in the development of FAS-targeted therapeutics, although this work must be considered as a preliminary investigation. Other and more in-depth future evaluations will be necessary to improve the MA structural features and better understand its selectivity of action and biocompatibility.

## 4. Materials and Methods

### 4.1. Myrianthic Acid

MA was provided by the Department of Pharmacy at the University of Naples [9].

### 4.2. MA Target Identification through DARTS

HeLa cells were grown at 37 °C in 5% CO_2_ atmosphere, in Dulbecco’s modified Eagle medium with 10% (*v*/*v*) fetal bovine serum albumin, 100 U/mL penicillin and 100 mg/mL streptomycin (Sigma-Aldrich, St. Louis, MO, USA) and collected by centrifugation (1000× *g*, 5 min).

HeLa cells were suspended in PBS (137 mM NaCl, 2.7 mM KCl, 10 mM Na_2_HPO_4_, 2 mM KH_2_PO_4_, pH 7.4) supplemented with 0.1% *v*/*v* Igepal and proteases inhibitors cocktail (GeneSpin, Milan, Italy). They were mechanically lysed with a Dounce homogenizer at 4 °C. This obtained suspension was centrifugated at 10,000× *g* for 5 min at 4 °C to remove cellular debris. The supernatant was collected, and protein concentration was determined by means of Bradford assay (BioRad Laboratories, Hercules, CA, USA) and then diluted to 3 mg/mL, adding PBS. Lysate was split into four aliquots of equal volume (100 μL each). Three of these were incubated with different concentrations of myrianthic acid (1 μM, 10 μM, 100 μM) dissolved in DMSO; the fourth was incubated with DMSO as a control. After 1 h of incubation at room temperature under continuous shaking (Mini-Rotator, Biosan, Warren, MI, USA), samples were split again and treated with different concentrations of subtilisin (enzyme to proteins ratio of 1:2500, 1:1500 and 1:500 *w*/*w*) for 30 min at 25 °C under continuous shaking (500 rpm, Thermomixer, Biosan). Moreover, proteolysis was simulated on the control sample by adding an equal volume of water. The protease was quenched by adding PMSF 1 mM final concentration (phenylmethylsulphonyl fluoride, Sigma-Aldrich, St. Louis, USA) to each sample and shaken for 10 min, 25 °C 500 rpm. This experiment was carried out in triplicate.

For the electrophoretic separation, 7 μL of each sample was added to SDS-PAGE loading buffer (60 mM Tris/HCl pH 6.8, 2% SDS, 0.001% bromophenol blue, 10% glycerol, 2% 2-mercaptoethanol) and heated at 95 °C for 5 min. These mixtures were loaded on a 4–12% Bis-Tris CriterionTM XT Precast Gel (BioRad Laboratories), and electrophoresis was performed in BioRad equipment (BioRad Laboratories). The gel was then fixed for 15 min (fixing solution: 50% H_2_O, 40% MeOH, 10% AcOH), washed three times (10 min each) with H_2_O and then submitted to Coomassie staining (BioRad Laboratories) for 1 h at room temperature under continuous shaking. The excess dye was removed by extensively washing the gel with H_2_O and a scan image of the resulting gel was then obtained through LabScan. Bands whose intensity was correlated to myrianthic acid concentration were submitted to in situ digestion protocol with trypsin [45]. Band pieces were washed by shrinking/swelling cycles using CH_3_CN and ammonium bicarbonate (AmBic, 50 mM, pH 8.5), alternatively. Then, disulphide bonds were reduced by treating the gel pieces with 1,4-dithiothreitol (DTT, 6.5 mM in 50 mM AmBic, 60 min, 60 °C) and the formed thiols were carboxyamidomethylated with iodoacetamide (IAA, 54 mM in 50 mM AmBic, 30 min, room temperature, in the dark). Residual reagents were removed by shrinking/swelling cycles, and gel pieces were rehydrated in a 12 ng/µL trypsin/LysC solution (Promega, Madison, WI, USA) on ice for 1 h. The excess enzymes were then removed, and 50 µL of 50 mM AmBic were added to allow protein digestion to proceed overnight at 37 °C (Thermomixer, Eppendorf, Hamburg, Germany). The supernatant was then collected, and peptides were extracted from the gel slices, shrinking them twice with 100% CH_3_CN. All the supernatants were collected and combined to be then dried out under vacuum (SpeedVac Concentrator Plus, Eppendorf) and solubilized in 30 µL of 10% Formic Acid (FA) for the subsequent nano-flow RP-UPLC MS/MS analysis. An amount of 1 µL of each sample was submitted to nano-flow RP-UPLC MS/MS analysis performed on the Orbitrap Q-Exactive Classic Mass Spectrometer (ThermoFisher Scientific, Bremen, Germany) coupled to an UltiMate 3000 Ultra-High Pressure Liquid Chromatography (UPLC) system (ThermoFisher Scientific, Bremen), equipped with an EASY-Spray PepMAPTM RSLC C18 column (3 μm, 100 Å, 75 μm × 50 cm, ThermoFisher Scientific, Bremen). Gradient ranged from 1 min at 3% B up to 95% of B in 60 min (A: 95% H_2_O, 5% CH_3_CN, 0.1% Acetic Acid (AA); B: 95% CH_3_CN, 5% H_2_O, 0.1% AA. ESI source parameters were set as follows: capillary temperature 250 °C; sheath and auxiliary gas flow (N_2_) 0 (arbitrary units); AGC target 3 × 10^6^; maximum IT 50 ms. MS spectra were acquired by full range acquisition covering *m*/*z* 375–1500 with a resolving power of 70,000. To obtain their HRMS/MS product ions, a data-dependent scan experiment was performed with the following parameters: loop count 8; resolution 17,500; AGC target 1 × 10^5^; maximum IT 80 ms; normalization collision energy at 26%, isolation width at 1.6.

Obtained raw files were uploaded into Proteome Discoverer (Proteome Discoverer 2.4^TM^ Software, ThermoFisher Scientific, Bremen), which used in silico SwissProt and Sequest HT with multi-peptide search and percolator validation data for the protein’s identification (parameters setting: maximum of two missed cleavages, trypsin digestion, fixed modification: carbamidomethyl (C); variable modifications: oxidization (M) and protein N-terminal acetylation.

A semi-quantitative analysis was also carried out, which led to the protein abundance in each sample treated with MA compared to the relative control. Data analysis was conducted as follows: proteins with abundance ratio “lysate/ctrl” <2 were excluded (“lysate” = sample with no MA and no subtilisin, “ctrl” = sample with no MA and with subtilisin). The resulting list was filtered, considering as protected only the proteins with increasing abundances according to MA concentrations.

The mass spectrometry proteomics data have been deposited to the ProteomeXchange Consortium via the PRIDE [46] partner repository with the dataset identifier PXD052306.

### 4.3. Western Blotting for DARTS Validation

An amount of 7 μL of each DARTS sample was treated with Laemmli buffer and heated at 95 °C for 5 min for the electrophoresis separation in an 8% SDS-PAGE gel. Proteins were then blotted onto nitrocellulose membrane, which was then dipped in a 5% non-fat dried milk containing T-TBS solution (31 mM Tris pH 8, 170 mM NaCl, 3.35 mM KCl, 0.05% tween 20 5%) for 1 h at room temperature under continuous shaking and then incubated overnight at 4 °C in agitation with the primary mouse antibody against the protein fatty acid synthase (FAS) (1:500 *v*/*v*, Santa Cruz Biotechnology, Dallas, TX, USA). The membrane was then washed three times with T-TBS to remove the excess antibody and re-incubated with a mouse peroxidase-conjugated secondary antibody (1:2500 *v*/*v*, Thermo-Scientific) at room temperature for 1 h under shaking. Signal was developed thanks to an enhancer solution combined with a peroxide solution (GeneSpin), and the signal was detected with LAS 4000 (GE Healthcare Waukesha, WI, USA). The procedure was repeated, incubating the membrane with the primary mouse anti-Glyceraldehyde 3-Phosphate Dehydrogenase antibody (GAPDH, 1:2500 *v*/*v*, Santa Cruz Biotechnology) to avoid errors due to the gel loading.

### 4.4. MRM Methods Fine-Tuning

The human build of the proteomic data repository PeptideAtlas was used to select FAS (UniProt Accession: P49327) tryptic peptides, which were in turn researched in SRMAtlas to identify their daughter ions. The SRMAtlas query parameters were set as follows: number of highest intensity fragment ions to keep: 8; target instrument: QTRAP 5500; transitions source: QTOF, Agilent QQQ, Qtrap5500, Ion Trap, Predicted; precursor exclusion range: kept blank; search proteins form: SwissProt; duplicate peptides: unique in results; heavy label: kept blank; labelled transitions: kept as default; maximum *m*/*z*: 1100 Da; minimum *m*/*z*: kept blank; allowed ions types: b-ions and y-ions; allowed peptide modification: carbamidomethylation of cysteines (C[160]).

Thus, the three best transitions for each peptide were selected to build MRM methods that were subsequentially tested on a tryptic-digested HeLa cell lysate treated as already described [47]. The final mixture of peptides was then dissolved in 10% FA to have a final peptide concentration of 2 µg/µL. For the chromatographic separation 15 µL were injected and separated on a Kinetex PS C18 column (50 × 2.1 mm, 2.6 μm, 100 Å, Phenomenex, Torrance, CA, USA) using a gradient from 5% to 95% of B in 20 min (A: 0.1% FA in H_2_O, B: 0.1% FA in CH_3_CN).

The 6500 Q-Trap was configured for ion spray operation, and the peptides were detected using multiple reaction monitoring (MRM) in positive ion mode. Values of additional QTrap parameters were as follows: curtain gas (CUR) = 30; ion-spray voltage (IS) = 5500; temperature (TEM) = 250; ion source gas 1 (GS1) = 25; ion source gas 2 (GS2) = 25; declustering potential (DP) = 80, entrance potential (EP) = 15, collision cell exit potential (CXP)= 12. The total cycle time was set to have at least 11 points across all chromatographic peaks. Data acquisition and processing were performed using Analyst software 1.6.2 (ABSciex, Foster City, CA, USA).

### 4.5. T-LiP-MRM-MS for the Interactions Study of the Complexes MA-FAS

HeLa cell lysate, obtained as previously described, was incubated with MA 100 μM and with DMSO for 1 h at room temperature under shaking (Mini-Rotator, Biosan), and then treated with subtilisin (enzyme to proteins ratio of 1:1500) for 30 min at 25 °C under continuous shaking (500 rpm, Thermomixer, Biosan), leaving an undigested sample (with no MA) as a control. Once the enzyme had been quenched with PMSF (1 mM final concentration), samples were first submitted to denaturant conditions with urea 4 M and then to the in-solution digestion protocol. Thus, MRM-MS analysis was performed in triplicate using the FAS MRM methods already optimized (as described in the previous paragraph). Analyst Software (AB Sciex) was used to measure the areas of each tryptic peptide peak.

### 4.6. Molecular Docking

The crystallographic structure of MAT domain of FAS with PDB code: 5MY0 [30] was used for docking calculations. This FAS structure is in complex with Malonyl CoA, the latter used as reference for grid box generation. The complex was prepared using the Protein Preparation Wizard at a pH of 7.4 ± 1.0, adding missing hydrogen atoms, and assigning bond orders. Finally, all water molecules were removed.

The grid used for all calculations featured innerbox dimensions of 10 Å and outerbox dimensions of 31.46 Å (center coordinates in Å: X = 30.53; Y = 288.57; Z = −158.52) to cover the active site region.

The structures of ursolic and myrianthic acids were drawn with 2D sketcher of Maestro (version 12.7, Schrödinger Suite, LLC, New York, NY, USA, 2021), in the Schrödinger Suite and were prepared using LigPrep module (version 5.7, Schrödinger, LLC, New York, NY, 2021), accounting for a pH = 7.4 ± 1.0 for the protonation state and minimizing the structure with OPLS 2005 force field, retaining the specified chirality and generating all the possible tautomers.

Docking calculations were performed using as input the prepared structures of ursolic and myrianthic acids in Glide software (version 9.0, Schrödinger, LLC, New York, NY, USA, 2021) at extra precision (XP) level using the Ligand docking panel in Schrödinger Suite. Specifically, 20 poses for ligand were saved in order to perform an exhaustive sampling of all possible conformations in the binding site of MAT domain of FAS binding site. The selection of the best docking poses of the investigated compounds was carried out through visual inspection based on both the docking score values and the establishment of interactions with fundamental amino acids.

### 4.7. Biological Evaluations

#### 4.7.1. Cancer Cell Cultures

Epithelial-like type human triple-negative breast adenocarcinoma cells MDA-MB-231 (ATCC, HTB-26TM) and epithelial-like human estrogen responsive (ER) breast adenocarcinoma cells MCF-7 (ATCC, HTB-22™) were grown in DMEM (Invitrogen, Paisley, UK) supplemented with 10% fetal bovine serum (FBS, Cambrex, Verviers, Belgium), L-glutamine (2 mM, Sigma, Milan, Italy), penicillin (100 units/mL, Sigma) and streptomycin (100 μg/mL, Sigma), and cultured in a humidified 5% carbon dioxide atmosphere at 37 °C, according to ATCC recommendations [48].

#### 4.7.2. Healthy Cell Cultures

Human primary adult dermal fibroblasts (HDFa) and human normal breast MCF-10A (ATCC, CRL-10317TM) cell lines were obtained from the American Type Culture Collection (ATCC; Virginia, VA, USA) and used as control healthy cells providing ideal cell systems to study toxicological cellular responses [39]. Cells were cultured in a humidified 5% carbon dioxide atmosphere at 37 °C, according to ATCC’s recommendations. Fibroblasts were cultured in Fibroblast Basal Medium (ATCC) supplemented with recombinant human fibroblast growth factor (rh FGF, 5 ng/mL), L-glutamine (7.5 mM), ascorbic acid (50 µg/mL), hydrocortisone hemisuccinate (1 µg/mL), rh Insulin (5 µg/mL) and Fetal Bovine Serum (FBS, 2%). Moreover, Penicillin-Streptomycin-Amphotericin B Solution (Penicillin: 100 Units/mL, Streptomycin: 100 µg/mL, Amphotericin B: 25 ng/mL) was added. HDFa cells were seeded at a density between 2.5–5 × 103 cells/cm^2^ and were passed when approximately 80% to 100% confluence was reached and only if cells were actively proliferating. Breast epithelial normal cells were cultured at 37 °C and 5% CO_2_ in DMEM-F12 medium supplemented with 10% fetal bovine serum, 100 μg/mL epidermal growth factor (EGF), 1 mg/mL hydrocortisone, 10 mg/mL insulin, 100 U/mL penicillin G and 100 μg/mL streptomycin.

#### 4.7.3. Bioscreens In Vitro and IC_50_ Evaluation

Bioactivity and cell responses to in vitro treatment with compounds under investigation were investigated through the estimation of a “cell survival index”, arising from the combination of cell viability evaluation with cell counting. The cell survival index was calculated as the arithmetic mean between the percentage values derived from the MTT assay and the automated cell count [49]. MDA-MB-231, MCF-7, MCF-10A, and HDFa cells were inoculated in 96-microwell culture plates at a density of 104 cells/well and allowed to grow for 24 h. Subsequently, the culture medium was exchanged with a fresh medium, and cells were exposed to varying concentrations (5→50 μM) of myrianthic acid (MA), ursolic acid (UA), and Orlistat for an additional 48 and 72 h. After the treatments, the medium was removed, and the cells were incubated with 20 μL/well of a MTT solution (5 mg/mL MTT, Sigma) for 1 h in a humidified 5% CO_2_ incubator at 37 °C. The incubation was stopped by removing the MTT solution and by adding 100 μL/well of DMSO to solubilize the obtained formazan. Finally, the absorbance was monitored at 550 nm using a microplate reader (iMark microplate reader, Bio-Rad, Milan, Italy). Cell number was determined by TC20 automated cell counter (Bio-Rad, Milan, Italy), which uses disposable slides, TC20 trypan blue dye (0.4% trypan blue dye w/v in 0.81% sodium chloride and 0.06% potassium phosphate dibasic solution) and a CCD camera to count cells based on the analyses of captured images. The medium was removed, and the cells were collected. Ten microliters of cell suspension, mixed with 0.4% trypan blue solution at 1:1 ratio, were loaded into the chambers of disposable slides. The results are expressed in terms of total cell count (cell number/mL). If the presence of trypan blue is identified, the instrument incorporates the dilution factor and displays both the count of live cells and the percentage of viability. Total counts and live/dead ratio from random samples for each cell line were subjected to comparisons with manual hemocytometers in control experiments.

The determination of the IC_50_ relies on plots of data (*n* = 6 for each experiment), repeated five times for a total of 30 samples (*n* = 30). Concentration–effect curves were obtained with nonlinear regression using GraphPad Prism 8.0 curve-fitting program [49].

#### 4.7.4. Preparation of Cellular Extracts

MDA-MB-231 and MCF-7 cells were harvested by treatment with trypsin-EDTA solution, pelleted by centrifugation, washed twice, and resuspended in PBS. Then, cells were sonicated for 30 min at 4 °C (SONICS VibracellTM) in PBS (pH 7.4) containing protease inhibitors (Roche) and centrifuged at 14,000× *g* for 15 min at 4 °C to obtain particle-free supernatants as soluble fraction stored at −80 °C. Supernatant samples were appropriately processed to measure protein content using the Bio-Rad assay (Bio-Rad Laboratories, Milan, Italy).

#### 4.7.5. Evaluation of FAS Activity in Protein Extracts

Adequate aliquots from cellular extracts of MDA-MB-231 and MCF-7 cells were diluted to a final concentration of 1 μg/μL. One hundred and twenty microliters of this particle-free supernatant were preincubated for 15 min at 37 °C for temperature equilibration. The sample was then added to 150 μL of the reaction buffer mixture [200 mmol/L potassium phosphate buffer (pH 7.0), 1 mmol/L EDTA, 1 mmol/L DTT, 30 μmol/L acetyl-CoA, 0.24 mmol/L NADPH, and 50 μmol/L malonyl-CoA], followed by 30 μL of 500 μmol/L malonyl-CoA (as the FAS substrate). A final volume of 0.3 mL of reaction was spectrophotometrically (340 nm) assayed for 10 min to evaluate FAS-dependent oxidation of NADPH. Before the addition of malonyl-CoA, the background rate of NADPH oxidation in the presence of acetyl-CoA was monitored at 340 nm for 3 min. FAS activity was expressed in nmol NADPH oxidized × min^−1^ × mg protein^−1^. The results are reported as percentage (%) of total FAS activity in the presence of 10 µM of Orlistat, ursolic acid, and myrianthic acid, respectively. In the absence of molecules under consideration, 100% was attributed to FAS activity. EDTA, dithiothreitol, acetyl-CoA, malonyl-CoA, and NADPH were purchased from Sigma.

#### 4.7.6. Evaluation of FAS Activity in Cells

MDA-MB-231 and MCF-7 cells were cultured under the experimental conditions specified above in a sterile Falcon^®^ 6-well Clear Flat Bottom TC-treated Multiwell Cell Culture Plate. Cells were inoculated at a density of 10^4^ cells/well and allowed to grow for further 24 h. After that, cells were exposed for 48 h to myrianthic acid MA, ursolic acid (UA), and Orlistat at a concentration of 10 µM. At the end of the experiment, cells were appropriately collected by trypsin-EDTA solution and then lysed to obtain cell extracts (see above). Finally, cell extracts were processed as described to evaluate the enzymatic activity of FAS.

#### 4.7.7. Western Blot Analysis

For Western blot analysis, samples containing 30 µg of proteins from MDA-MB-231 cell lysates were loaded on 8% SDS–PAGE and transferred to nitrocellulose membranes. After blocking at room temperature in milk buffer [1 × PBS, 5–10% (*w*/*v*) non-fat dry milk, 0.2% (*v*/*v*) Tween-20], the membranes were incubated at 4 °C overnight with the primary mouse antibody against the protein fatty acid synthase (FAS) (1:500 *v*/*v*, Santa Cruz Biotechnology). The membrane was then washed three times with T-TBS to remove the excess antibody and re-incubated with a mouse peroxidase-conjugated secondary antibody (1:2500 *v*/*v*, Thermo-Scientific). The resulting immunocomplexes were visualized by the ECL chemoluminescence method (ECL, Elabscience, Tucson, AZ, USA) and analyzed by an imaging system (ChemiDoc, Bio-Rad). Densitometric analysis was conducted using ImageJ software. Normalization of results was ensured by incubating the nitrocellulose membranes in parallel with the β-actin antibody (Sigma-Aldrich, 1:2500 *v*/*v*) [50].

## 5. Conclusions

This study highlights the promising role that myrianthic acid (MA) can play in drug discovery by revealing a new mechanism of action that can be leveraged in the search for novel therapeutic compounds. Understanding the comparative effectiveness of MA is crucial in assessing its feasibility for therapeutic applications. Indeed, the exploration of MA as a potential FAS inhibitor opens new avenues for addressing conditions associated with dysregulated FAS activity. As with the structural analogue ursolic acid (UA), MA ability to also reduce cellular expression of FAS makes it particularly attractive for the xenobiotic control of lipid metabolism. In particular, this study showcases that there is potential to develop MA into an anticancer therapeutic for the management of lipogenic tumor phenotypes overexpressing FAS protein.

## Figures and Tables

**Figure 1 ijms-25-05918-f001:**
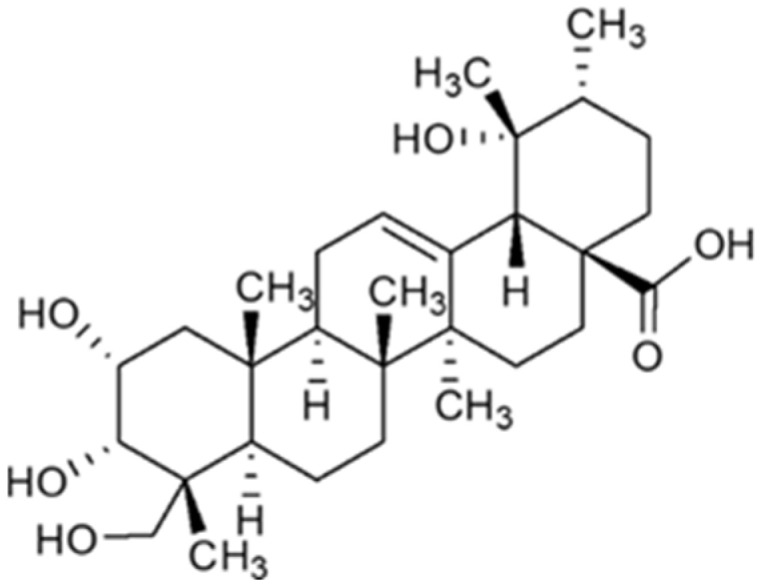
Chemical structure of Myrianthic acid (MA).

**Figure 2 ijms-25-05918-f002:**
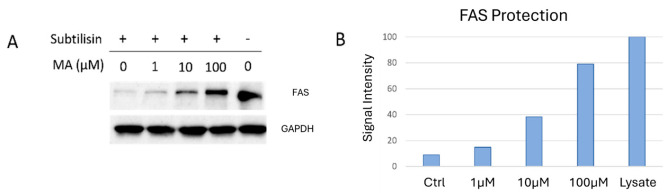
(**A**) Western blotting analysis carried out on the DARTS samples reveals FAS protection upon MA interaction. GAPDH was used as a loading normalizer. Western blotting densitometric analysis for FAS (**B**) was performed through ImageJ. Undigested proteins (i.e., Lysate sample) were rated as 100%.

**Figure 3 ijms-25-05918-f003:**
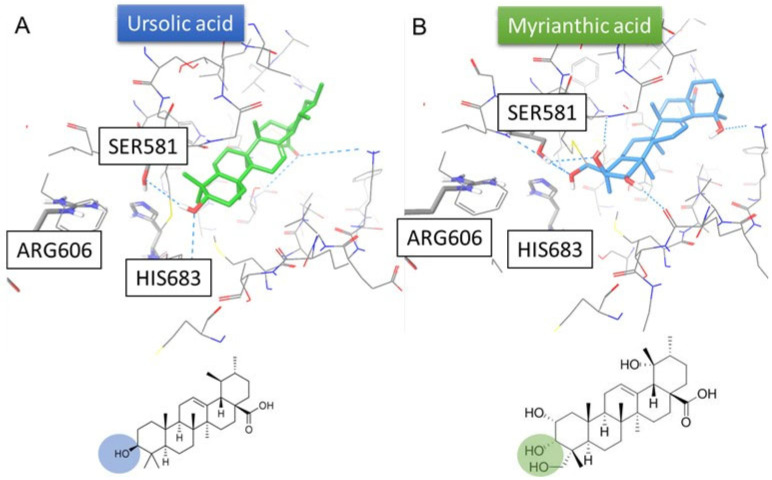
Predicted binding mode according to the first hypothesis of ursolic acid (panel A, colored by atom type: C green, O red, polar H white) and MA (panel B, colored by atom type: C light blue, O red, polar H white) in the MAT domain binding site (PDB: 5MY0, key residues are reported as sticks and colored by atom type: C grey, O red, N blue, S yellow, polar H light grey). The hydroxyl groups interacting with the catalytic triad are highlighted in blue for ursolic acid (**A**) and green for MA (**B**). H-bonds are represented by cyan dotted lines.

**Figure 4 ijms-25-05918-f004:**
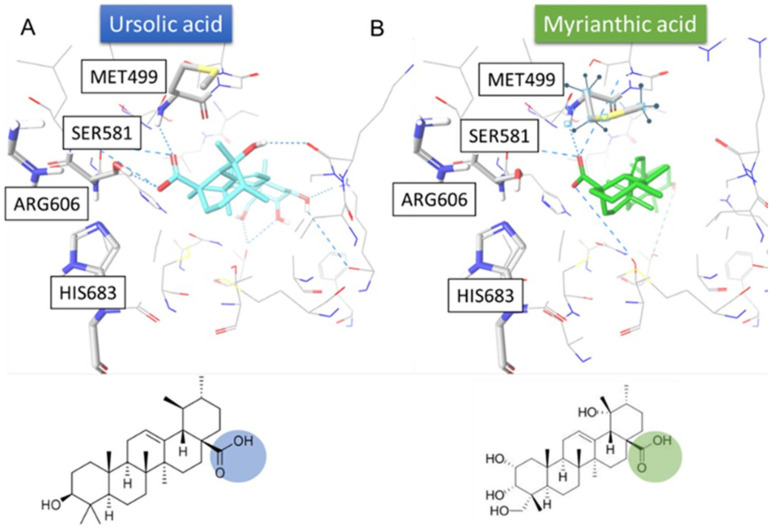
Predicted binding mode according to the second hypothesis of ursolic acid (panel A, colored by atom type: C cyan, O red, polar H white) and MA (panel B, colored by atom type: C green, O red, polar H white) in the MAT domain binding site (PDB: 5MY0, key residues are reported as sticks and colored by atom type: C grey, O red, N blue, S yellow, polar H light grey). The carboxylic groups interacting with the catalytic triad are highlighted in blue for ursolic acid (**A**) and green for MA (**B**). H-bonds are represented by cyan dotted lines.

**Figure 5 ijms-25-05918-f005:**
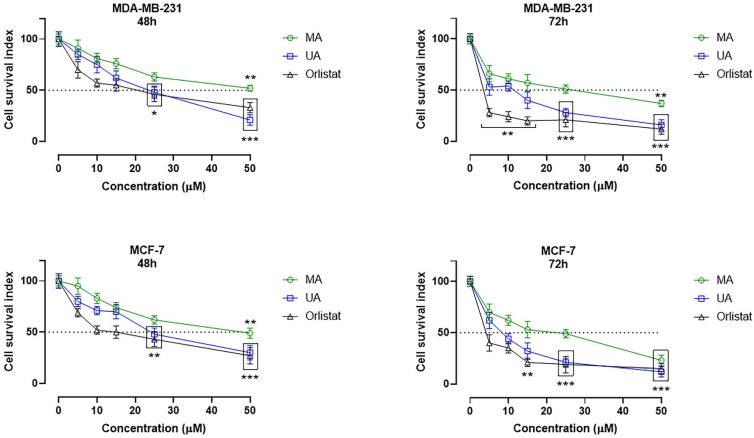
Cell survival index, evaluated by the MTT assay and live/dead cell ratio, for MDA-MB-231 and MCF-7 BC cell lines following 48 and 72 h of incubation with the indicated concentrations (0–50 µM) of myrianthic acid (MA), ursolic acid (UA), and Orlistat, as indicated in the legend. Data are expressed as a percentage of untreated control cells and are reported as the mean of five independent experiments ± SEM (*n* = 30). The cell survival index was calculated as described in the experimental section and plotted in line graphs against the different concentrations of the tested molecules. * *p* ˂ 0.05 vs. control cells; ** *p* < 0.01 vs. control cells; *** *p* < 0.001 vs. control cells.

**Figure 6 ijms-25-05918-f006:**
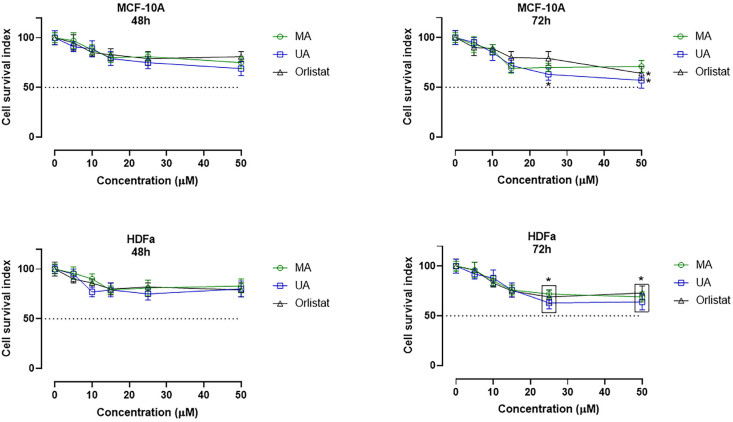
Cell survival index, evaluated by the MTT assay and live/dead cell ratio, for MCF-10A and HDFa following 48 and 72 h of incubation with the indicated concentrations (0–50 µM) of myrianthic acid (MA), ursolic acid (UA), and Orlistat, as indicated in the legend. Data are expressed as a percentage of untreated control cells and are reported as the mean of five independent experiments ± SEM (*n* = 30). The cell survival index was calculated as described in the experimental section and plotted in line-graphs against the different concentrations of the tested molecules. * *p* ˂ 0.05 vs. control cells.

**Figure 7 ijms-25-05918-f007:**
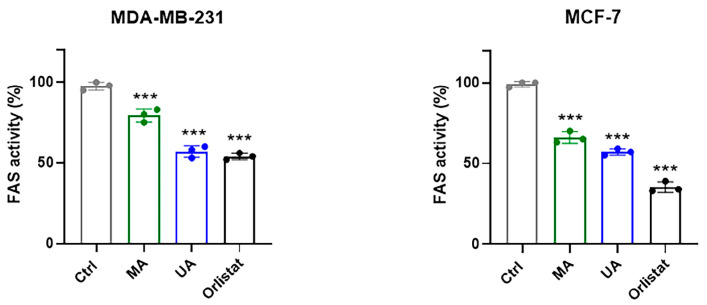
Bar graphs encompassing representative data from the FAS activity assay in tumor cell lysates obtained from the indicated cell lines by monitoring the oxidation of NADPH, as described in the experimental section. In the absence of molecules under consideration, 100% was attributed to FAS activity. Results are expressed as percentage (%) of total FAS activity in presence of 10 µM of MA, UA, and Orlistat, respectively, and are reported as mean of three independent experiments ± SEM (*n* = 15). *** *p* < 0.001 vs. untreated controls.

**Figure 8 ijms-25-05918-f008:**
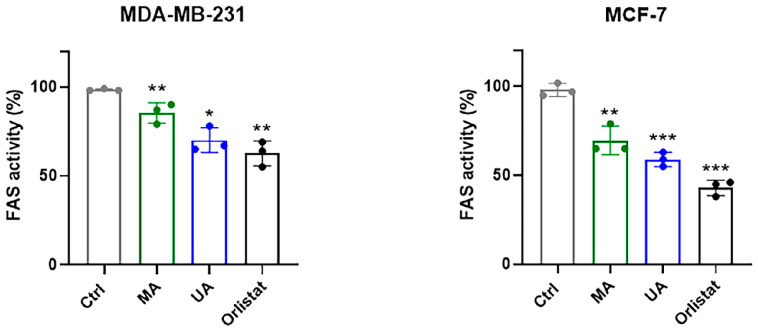
Bar graphs relative to the FAS activity assay in the indicated BC cells protein extracts by monitoring the oxidation of NADPH, as described in the experimental section. Cells were treated in vitro for 48 h with 10 µM of MA, UA, and Orlistat, as indicated in the legend, and then appropriately lysed to obtain cellular extracts. FAS activity in lysates from untreated control cells had 100% attributed to it. Results are expressed as percentage of FAS activity and are reported as mean of three independent experiments ± SEM (*n* = 15). * *p* ˂ 0.05 vs. control cells; ** *p* < 0.01 vs. control cells; *** *p* < 0.001 vs. control cells.

**Figure 9 ijms-25-05918-f009:**
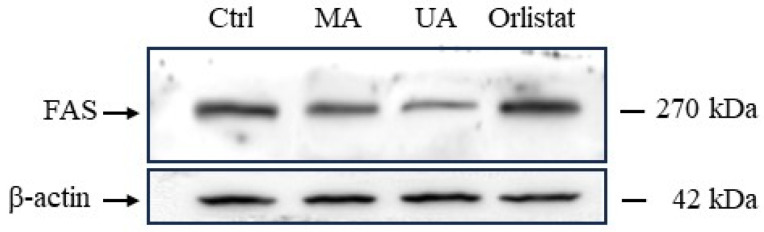
Western blot analysis showing the effects of 10 µM concentrations of myrianthic acid (MA), ursolic acid (UA), and Orlistat following 48 h of incubations in MDA-MB-231 cells on the expression of FAS. The shown blots are representative of three independent experiments and are cropped from different parts of the same gel (provided in Appendix A), as explicit by using clear delineation with dividing lines and white space.

**Table 1 ijms-25-05918-t001:** PD protection values across replicates for FAS expressed as ratios between the abundance in samples treated with different MA amounts and the control sample. Lys/ctrl is the abundance ratio across the positive control (no MA, undigested) and the negative control (no MA, digested).

Fatty Acid Synthase Abundance				
	1 µM/ctrl	10 µM/ctrl	100 µM/ctrl	Lysate/ctrl
Replicate A	1.829	2.594	3.962	15.79
Replicate B	0.431	0.518	2.114	5.966
Replicate C	2.517	3.368	5.443	23.426

**Table 2 ijms-25-05918-t002:** Protected FAS peptides. For each peptide reported its Q1 and Q3 values, the ID and the fold change values for the treated and the control sample with the *p*-value.

			Treated		Lysate	
Q1_mz	Q3_mz	ID	Fc	*p*-Value	Fc	*p*-Value
808.11	215.14	T[2483-2505]R	6.46	3.18 × 10^−3^	52.85	4.12 × 10^−4^
718.34	851.9	E[612-631]K	1.79	5.00 × 10^−2^	8.47	1.36 × 10^−2^
623.94	504.28	F[1117-1131]R	1.70	3.05 × 10^−2^	10.72	1.02 × 10^−2^
610.99	495.3	F[1772-1787]K	2.49	1.55 × 10^−2^	6.04	1.07 × 10^−2^
546.26	637.29	C[1828-1841]R	1.75	1.56 × 10^−3^	1.65	4.53 × 10^−2^
807.42	515.33	E[2207-2220]R	3.04	7.20 × 10^−3^	3.11	1.93 × 10^−2^
775.40	291.17	A[943-957]K	1.67	3.58 × 10^−3^	5.79	1.64 × 10^−3^
703.41	708.45	D[2126-2138]R	7.56	3.93 × 10^−3^	24.72	1.03 × 10^−2^
596.79	691.31	L[203-213]K	2.29	1.76 × 10^−2^	14.82	2.55 × 10^−3^
449.26	204.13	L[1583-1591]K	1.78	1.66 × 10^−3^	2.06	4.42 × 10^−3^

**Table 3 ijms-25-05918-t003:** IC_50_ values (µM) relative to MA, and to UA and Orlistat used as reference drugs. in the indicated breast cancer cell lines (MDA-MB-231 and MCF-7) and healthy cells (MCF-10A and HDFa) following 48 h of incubation. IC_50_ values are calculated from concentration-effect curves and reported as mean values ± SEM (*n* = 30).

IC_50_ (µM)—48 h				
	MDA-MB-231	MCF-7	MCF-10A	HDFa
Myrianthic acid	50 ± 1	46 ± 2	>50	>50
Ursolic acid	23 ± 5	24 ± 3	>50	>50
Orlistat	19 ± 3	13 ± 4	>50	>50

**Table 4 ijms-25-05918-t004:** IC_50_ values (µM) relative to MA, and to UA and Orlistat used as reference drugs. in the indicated breast cancer cell lines (MDA-MB-231 and MCF-7) and healthy cells (MCF-10A and HDFa) after 72 h of incubation. IC_50_ values are calculated from concentration-effect curves and reported as mean values ± SEM (*n* = 30).

IC_50_ (µM)—72 h				
	MDA-MB-231	MCF-7	MCF-10A	HDFa
Myrianthic acid	25 ± 3	21 ± 6	>50	>50
Ursolic acid	11 ± 3	9 ± 2	>50	>50
Orlistat	4 ± 2	5 ± 1	>50	>50

## Data Availability

All data are contained within the article and Appendix A. Data are available via ProteomeXchange with identifier PXD052306.

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
