# Peer review of "Fatty Acid Synthase as Interacting Anticancer Target of the Terpenoid Myrianthic Acid Disclosed by MS-Based Proteomics Approaches"

_ijms, 2024, doi:10.3390/ijms25115918_

Round 1
Reviewer 1 Report
Comments and Suggestions for Authors
Using MS-based chemical proteomic approaches, this study focuses on the target deconvolution of the natural triterpene molecule 19 myrianthic acid. Fatty acid synthase (FAS) was discovered to be an intriguing macro-22 molecular equivalent of myrianthic acid through the use of Drug Affinity Re-20 sponsive Target Stability (DARTS) and targeted-Limited Proteolysis linked to Mass Spectrometry 21 (t-LiP-MS). The data presented here is scientifically sound and the authors gathered enough follow up validation assays that support most of the scientific claims specifically the potential inhibitory activity of myrianthic acid against FAS enzyme and consequently resulted in the antiproliferative activity towards tumor cells. However this reviewer has major concerns regarding the current structure of the text and missed placed sections. Specifically:
Introduction - is too long and can be reduced by 2/3rds
Results Page (P) 3 Line (L) 113- 129. This part should be moved to introduction
P3 L130 -L166 this is part of M&M and should be removed from results section
Discussion: Overall, it is excessively long and could be substantially condensed to concentrate on the findings' discussion. The experiments conducted and the data obtained are described in P11 L387–L422, with almost no discussion of the findings. I propose that the authors revise and/or write the discussion section to accurately represent a critical evaluation of the findings by incorporating relevant literature.
Data should be make available through a online public data depository - E.g. PRIDE, ProteomeXexchange
Reviewer 2 Report
Comments and Suggestions for Authors
1,The researchers have detected the binding between MA and FAS through mass spectrometry-based proteomics methods. However, this multi-component omics test may result in false positive results, especially when the binding ability between the two is weak. Why didn't other in vitro experiments, such as ITC and SPR, be used to verify their interactions and the magnitude of their binding constants?
2, In further studying the details of the interaction between FAS and MA, the author cited existing MAT crystal structures and proposed two possible hypotheses using molecular docking techniques, which is a very good idea. But why not try the co-crystallization of FAS and MA? If successful, all problems can be fundamentally solved.
3, There are some minor errors and irregularities in the manuscript:
1)There are many formats in the manuscript that need to be modified, such as 1uM and 100uM in line 497, 100uM in line 598, and so on.
2)The "Usolic acid" in "interaction of both MA and Usolic acid" on line 238 should be FAS.
3)The statement "identity=87.07%" in line 243 seems somewhat non-standard.
Reviewer 3 Report
Comments and Suggestions for Authors
Summary:
The article determines by Drug Affinity Responsive Target Stability (DARTS) and targeted-Limited Proteolysis coupled to Mass Spectrometry (t-LiP-MS) that the enzyme fatty acid synthase (FAS) is an interesting macromolecular of myrianthic acid (MA) and it is confirmed by molecular docking. In this article, the authors also evaluated the inhibitory effect of FAS enzyme induced by (MA) through several biological assays in vitro in tumoral cell lines. Therefore, the authors highlight a possible new natural therapeutic compound for treating tumors by inhibiting FAS expression. Nevertheless, there are some comments to further improve the manuscript:
Major comments
1- In the current article, the abstract should begin with a general background about myrianthic acid.
2- The bioactivities of natural triterpenes such as the anticancer properties of myrianthic acid, are briefly mentioned in the introduction. It is highly recommended to mention other triterpenes with anticancer effects, especially the more recent ones.
3- In the discussion section the natural triterpenes are briefly mentioned, it is highly recommended to discuss the results obtained in this work with respect to other similar and recent studies done on natural triterpenes.
4- In the discussion, the authors should mention the limitations of their study and how they can enhance it in their future works.
Minor comments
5- The terms “in vitro” and “in vivo” should be written in italics.
6- The resolution of Figures 5 and 6 is poor. They should be improved.
Comments on the Quality of English LanguageThe English in the article is normally clear, and correctly structured, making it not difficult to follow the research findings and methodology. However, the article has minor grammatical errors.
Round 2
Reviewer 3 Report
Comments and Suggestions for Authors
The modifications introduced by the authors are satisfactory, depending on my major comments, but regarding the improvement of the resolution of Figures 5 and 6, is not satisfactory. The authors should improve their resolution because the resolution of both Figures 5 and 6 is very poor.
Author Response
The authors thank the reviewer. See attached file
